# Inducible depletion of adult skeletal muscle stem cells impairs the regeneration of neuromuscular junctions

Wenxuan Liu[1,2], Lan Wei-LaPierre[3], Alanna Klose[1], Robert T Dirksen[3], Joe V Chakkalakal[1,4,5]*

[1]Department of Orthopaedics and Rehabilitation, Center for Musculoskeletal Research, University of Rochester Medical Center, Rochester, United States; [2]Department of Biomedical Genetics, Genetics, Genomics and Development Graduate Program, University of Rochester Medical Center, Rochester, United States; [3]Department of Pharmacology and Physiology, University of Rochester Medical Center, Rochester, United States; [4]Stem Cell and Regenerative Medicine Institute, University of Rochester Medical Center, Rochester, United States; [5]The Rochester Aging Research Center, University of Rochester Medical Center, Rochester, United States

**Abstract** Skeletal muscle maintenance depends on motor innervation at neuromuscular junctions (NMJs). Multiple mechanisms contribute to NMJ repair and maintenance; however muscle stem cells (satellite cells, SCs), are deemed to have little impact on these processes. Therefore, the applicability of SC studies to attenuate muscle loss due to NMJ deterioration as observed in neuromuscular diseases and aging is ambiguous. We employed mice with an inducible Cre, and conditionally expressed DTA to deplete or GFP to track SCs. We found SC depletion exacerbated muscle atrophy and type transitions connected to neuromuscular disruption. Also, elevated fibrosis and further declines in force generation were specific to SC depletion and neuromuscular disruption. Fate analysis revealed SC activity near regenerating NMJs. Moreover, SC depletion aggravated deficits in reinnervation and post-synaptic morphology at regenerating NMJs. Therefore, our results propose a mechanism whereby further NMJ and skeletal muscle decline ensues upon SC depletion and neuromuscular disruption.

*For correspondence: joe_chakkalakal@urmc.rochester.edu

**Competing interests:** The authors declare that no competing interests exist.

## Introduction

Skeletal muscle is composed of long multinucleated cells, muscle fibers (myofibers), which function as primary effectors for force production and contribute to the regulation of whole body metabolism. Although primarily a post-mitotic tissue, adult skeletal muscle possesses a remarkable capacity for regeneration. This capacity is endowed by a population of resident stem cells, satellite cells (SCs), identified by the expression of the paired box transcription factor Pax7 (*Tajbakhsh, 2009*; *Brack and Rando, 2012*; *Yin et al., 2013*). In adults, SCs normally reside in a quiescent state at the interface between the myofiber and overlying basal lamina (*Yin et al., 2013*). However, in response to degenerative stimuli, Pax7+ SCs activate and divide to produce myogenic progenitors for skeletal muscle regeneration (*Tajbakhsh, 2009*; *Yin et al., 2013*). Since the initial identification of SCs over 50 years ago, many studies have examined roles for these cells in a plethora of distinct models of skeletal muscle injury, adaptability and disease (*Mauro, 1961*; *Relaix and Zammit, 2012*). Both depletion of Pax7+ SCs and targeted disruption of Pax7 have shown these cells to be an essential source of myonuclei for skeletal muscle regeneration (*Relaix and Zammit, 2012*; *Gunther et al., 2013*; *von Maltzahn et al., 2013*; *Lepper et al., 2011*; *Murphy et al., 2011*). Depletion studies have revealed roles for Pax7+ SCs in late stages of experimentally induced skeletal muscle hypertrophy (*Fry et al., 2014*).

**eLife digest** New muscle fibers are made throughout our lives to replace those that have been damaged by normal wear and tear, and to meet new physical demands. These new muscle fibers develop from a pool of muscle stem cells. To create and maintain fully working muscles, nerve cells called motor neurons must also properly attach to the muscle fibers. These nerve cells transmit messages from the brain that tell the muscles what to do. If the muscle-nerve connections do not form correctly, or are severed, muscles can waste away. This may occur as part of a neuromuscular disease, and also happens to some extent as a normal part of aging.

It was thought that muscle stem cells do not affect how the muscle-nerve connections form. By studying genetically engineered mice, Liu et al. now show that this is not the case. These mice had modifications to their muscle stem cells that allowed the number of these cells to be artificially reduced, and some cells also produced a fluorescent protein that allowed them to be tracked.

Surgically severing some of the muscle-nerve connections in the mice triggered the rebuilding of the connections, but also weakened the muscles and caused some disease-related changes in the muscle tissue. During the healing process, the muscle stem cells are active near the regenerating connections. Reducing the number of muscle stem cells in the mice while these broken connections were healing further weakened the muscles. Closer inspection of the muscle-nerve connections also revealed poorer quality connections were formed in the stem-cell deficient mice.

Further study of how stem cells help to form strong nerve-muscle connections may allow scientists to develop new treatments for age- or disease-related muscle loss.

Recently, SCs were also shown to function as a source of growth factors to facilitate bone fracture healing (*Abou-Khalil et al., 2015*). Similar strategies have shown SCs contribute to myofibers and regulate aspects of skeletal muscle integrity during aging (*Fry et al., 2015*; *Keefe et al., 2015*).

In adults, each myofiber is innervated by a single axon from a motor neuron (*Sanes and Lichtman, 1999*). Innervation occurs at a specialized site in the central region of myofibers, the neuromuscular junction (NMJ) (*Sanes and Lichtman, 1999*; *Wu et al., 2010*). The NMJ, which occupies approximately 0.1% of the surface area of a myofiber, initiates action potential propagation required for excitation/contraction coupling to generate force for movement and maintain myofiber properties (*Sanes and Lichtman, 1999*; *Bassel-Duby and Olson, 2006*; *Wu et al., 2010*; *Schiaffino and Reggiani, 2011*). Consistent with a vital role for NMJ integrity in skeletal muscle maintenance, neuromuscular disruptions elicit severe myofiber atrophy, and are frequently associated with skeletal muscle dysfunction observed in the context of neuromuscular diseases (NMDs) and aging (*Murray et al., 2010*; *Gonzalez-Freire et al., 2014*; *Moloney et al., 2014*).

The regeneration of NMJs in response to peripheral nerve lesions can occur, however the length and quality of recovery depends on the severity of injury (*Buti et al., 1996*; *Williams et al., 2009*). Remarkably, the initial reinnervation of synaptic basal lamina by motor axons can proceed in the absence of myofibers (*Sanes et al., 1978*; *Sanes and Lichtman, 1999*). However, in the absence of myofibers and associated SCs, the continued maintenance of reinnervated NMJs on basal lamina ghosts eventually declines (*Sanes et al., 1978*; *Sanes and Lichtman, 1999*). These observations indicate that myofiber derived factors or associated SCs may be required for the continued differentiation and maintenance of regenerated NMJs. Accordingly, myofiber components and derived factors have been identified to assist in the progressive differentiation of developing and regenerating NMJs (*Sanes and Lichtman, 1999*; *Fox et al., 2007*; *Williams et al., 2009*). In models of chronic denervation, where reinnervation is prevented, SCs activate and divide; however, the derived progenitors migrate into interstitial spaces, undergo defective differentiation or are lost via apoptosis (*Dedkov et al., 2001*; *Borisov et al., 2005*; *Bruusgaard and Gundersen, 2008*). Furthermore, little turnover of myonuclei and fusion of myogenic progenitors to chronically denervated parent myofibers have been observed (*Bruusgaard and Gundersen, 2008*). Collectively, these studies have suggested SCs have limited, if any, roles in the regeneration of NMJs upon neuromuscular disruptions (*Gundersen and Bruusgaard, 2008*).

Through the use of targeted genetic strategies we sought to reexamine the fates and roles of SCs in a model of peripheral nerve injury that enables NMJ regeneration. In this study we find a limited

proportion of SCs activate and divide during NMJ reestablishment. Remarkably, while SC depletion did not lead to additional loss of skeletal muscle mass, it was sufficient to reduce myofiber size, increase inter-myofiber connective tissue (MCT) accumulation, and aggravate myofiber type transitions connected to NMJ disruption. These phenotypes were associated with further declines in force generation capacity. Examination of fate revealed increased SC activity and fusion of indelibly labeled SC-derived progenitors to myofibers predominantly in the vicinity of regenerating NMJs. Consistent with a role for SCs in the regeneration of NMJs, we found that SC depletion led to deficits in NMJ reinnervation, reductions in post-synaptic morphology and loss of post-synaptic myonuclei. Collectively our findings reveal fates and roles for SCs in the regeneration of NMJs and regulation of skeletal muscle integrity upon neuromuscular disruption.

## Results

### SC depletion exacerbates myofiber atrophy and induces connective tissue accumulation in skeletal muscles after neuromuscular disruption

To examine the roles of SCs in skeletal muscles upon neuromuscular disruption we generated $Pax7^{CreER/+}$; $Rosa26^{DTA/+}$ (P7DTA) and $Pax7^{+/+}$; $Rosa26^{DTA/+}$ (Ctrl) mice. These mice enable tamoxifen (Tmx)-mediated expression of diphtheria toxin-A (DTA) to deplete Pax7+ SCs to levels that prevent the regeneration of skeletal muscle (*Murphy et al., 2011*; *Relaix and Zammit, 2012*). We employed 1-2 mm sciatic nerve transection (SNT) to disrupt lower-limb NMJs. This form of surgery leads to complete denervation of adult NMJs. Although delayed, reinnervation as assessed by immunofluorescence (IF) and physiological measures does occur 4–6 weeks after SNT (*Buti et al., 1996*; *Williams et al., 2009*). Consistent with previous reports, a modest albeit significant increase in Pax7+ SC number was observed 6 weeks after SNT surgery (*Figure 1*) (*Snow, 1983*; *Viguie et al., 1997*). After Tmx administration, extensive depletion of Pax7+ SCs occurred regardless of sham or SNT surgery (*Figure 1*).

We assessed various morphological parameters to determine the consequences of SC depletion on SNT-induced skeletal muscle atrophy. Gross examination of TA skeletal muscles 6 weeks after SNT surgery revealed overall reductions in girth, which were not noticeably different between P7DTA and Ctrl (*Figure 2A*). While the mass of TA, extensor digitorum longus (EDL) and soleus muscles 6 weeks after SNT relative to contralateral sham controls remained low, SC depletion did not induce further loss of muscle mass (*Figure 2B–D*). Examination of individual myofibers from EDL muscles that had been fixed prior to isolation from lower limbs revealed only modest reductions in myonuclear density after SNT and SC depletion (*Figure 2—figure supplement 1A and C*). Based on these moderate losses, a small albeit significant level of myonuclear turnover was observed in P7DTA sham muscles. Therefore, the magnitude of myonuclear loss after SNT relative to sham in P7DTA skeletal muscles is small (*Figure 2—figure supplement 1C*). Next we examined myofiber size based on Laminin IF analysis of transverse sections from TA muscles (*Figure 2E*). Despite 6 weeks of recovery, Ctrl myofiber size after SNT remained ~25% lower relative to contralateral sham, however, SC depletion led to further SNT-induced myofiber atrophy (*Figure 2F*). Distribution analysis of myofiber size did not reveal any significant differences between Ctrl and P7DTA muscles after sham surgery, but a significant shift towards smaller sizes after SNT surgery was observed upon SC depletion (*Figure 2G*). No change in myofiber numbers was observed after SNT and SC depletion (*Figure 2—figure supplement 1B*). Therefore, SC depletion did not lead to a significant change of overall muscle morphology and mass, but aggravated SNT-induced myofiber atrophy.

Previous studies on rodent models or human patients reveal that chronic denervation, aging of skeletal muscles or NMDs can induce an increase in MCT content, an indicator of fibrosis (*Peltonen et al., 1982*; *Savolainen et al., 1988*; *Goldspink et al., 1994*; *Brack et al., 2007*). Also, SC depletion has been shown to result in extracellular matrix accumulation in the context of skeletal muscle regeneration, functional overload-induced hypertrophy and aging (*Murphy et al., 2011*; *Fry et al., 2014*, *2015*). Therefore, to determine if elevated MCT is associated with reductions in myofiber size upon SC depletion and SNT, we performed hematoxylin and eosin (H&E) and Sirius Red staining for collagens (*Figure 3A,B*). Surprisingly, although SNT surgery alone did not increase MCT content, when combined with SC depletion, a significant increase in fibrosis was observed (*Figure 3C*). Therefore, the lack of change in the mass of SC-depleted skeletal muscles relative to Ctrl after SNT surgery was accompanied by both increased fibrosis and myofiber atrophy.

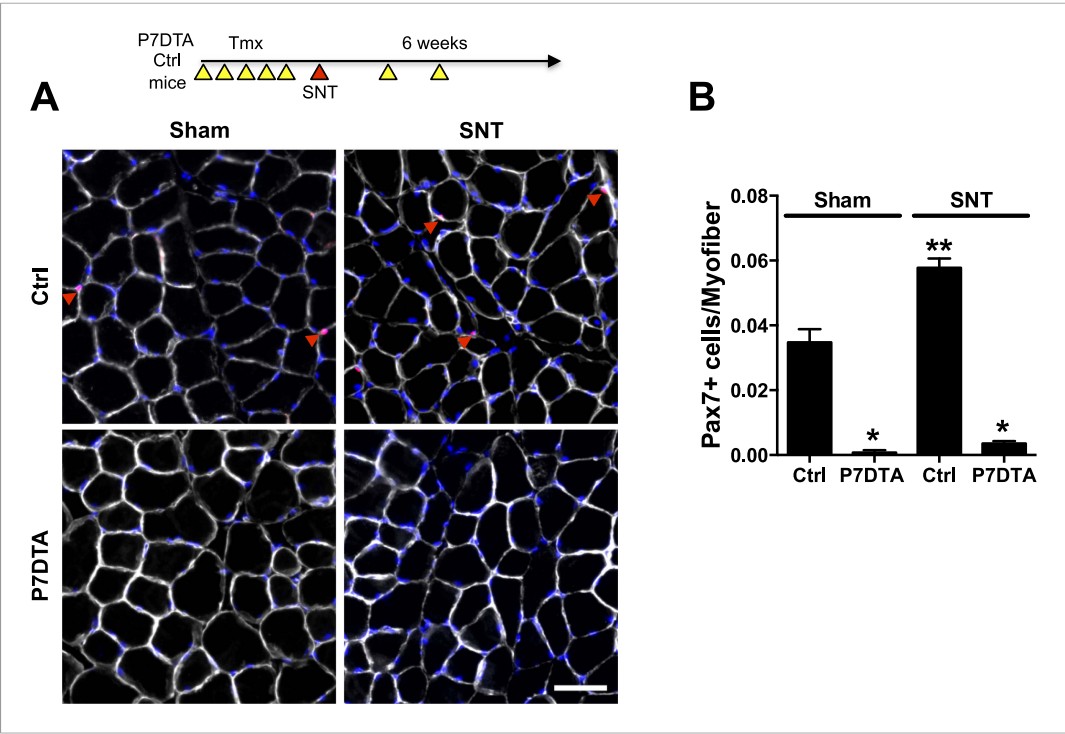

**Figure 1**. Depletion of Pax7+ SCs in P7DTA skeletal muscles. (**A**) Scheme demonstrating time of Tmx treatment, Sciatic nerve transection (SNT) surgery, and harvest of tissue. Representative images of TA transverse sections, stained with anti-Pax7 (red), anti-Laminin (white) and DAPI (blue). Red arrowheads indicate Pax7+ cells. (**B**) Quantification of Pax7+ satellite cell (SC) number from Ctrl and P7DTA TA muscles 6 weeks after sham or SNT surgery. N = 3 mice, 3 sections/mouse, 6 fields/ section. Scale bar = 50 μm. *p < 0.05 compared to Ctrl, **p < 0.05 compared to Ctrl sham, ANOVA/Bonferroni multiple comparisons test.

## SC depletion aggravates myofiber type transitions and functional deficits of skeletal muscles connected to neuromuscular disruption

Most skeletal muscles are composed of heterogeneous mixtures of functionally distinct types of myofibers that differ in many parameters (*Schiaffino and Reggiani, 2011*). Based on the expression of skeletal muscle myosin heavy chain (MyHC), different types of myofibers can be classified along a continuum: type I, IIA, IIX and IIB (*Schiaffino and Reggiani, 2011*). In this regard, type I fibers are characterized by slower contraction kinetics and lower force generation, whereas at the other end of the continuum type IIB fibers exhibit faster contraction kinetics and greater force generation (*Mantilla and Sieck, 2003*). Consistent with the vital influence of neural activity on myofiber type, myofibers innervated by the same motor neuron primarily express one MyHC isoform, and cross-innervation can induce the expression of isoforms indicative of the foreign nerve (*Murray et al., 2010*; *Moloney et al., 2014*). Neuromuscular disruption is frequently associated with abnormal profiles of MyHC isoform expression within a given skeletal muscle (*Klitgaard et al., 1990*; *Krivickas et al., 2002*; *Palencia et al., 2005*; *Stevens et al., 2008*; *Schiaffino and Reggiani, 2011*; *Rowan et al., 2012*). A typical feature of NMD and aging skeletal muscle is an increased occurrence of hybrid myofibers whereby two or more MyHC isoforms are co-expressed (*Klitgaard et al., 1990*; *Krivickas et al., 2002*; *Palencia et al., 2005*; *Stevens et al., 2008*; *Rowan et al., 2012*). To examine hybrid myofibers, we immuno-stained Ctrl and P7DTA inner TA/EDL muscle sections 6 weeks after sham or SNT surgery with antibodies that specifically detect MyHC IIX or all MyHCs except IIX, therefore hybrid IIX myofibers will be labeled with both. Very few MyHC IIX hybrid myofibers were found in adult inner TA/EDL muscles after sham surgery regardless of genotype (*Figure 4A,B*). Elevations in the proportion of hybrid myofibers were observed in Ctrl inner TA/EDL muscles 6 weeks after SNT, a phenotype exacerbated by SC depletion (*Figure 4A,B*). Assessment of inner TA/EDL muscles after SNT with

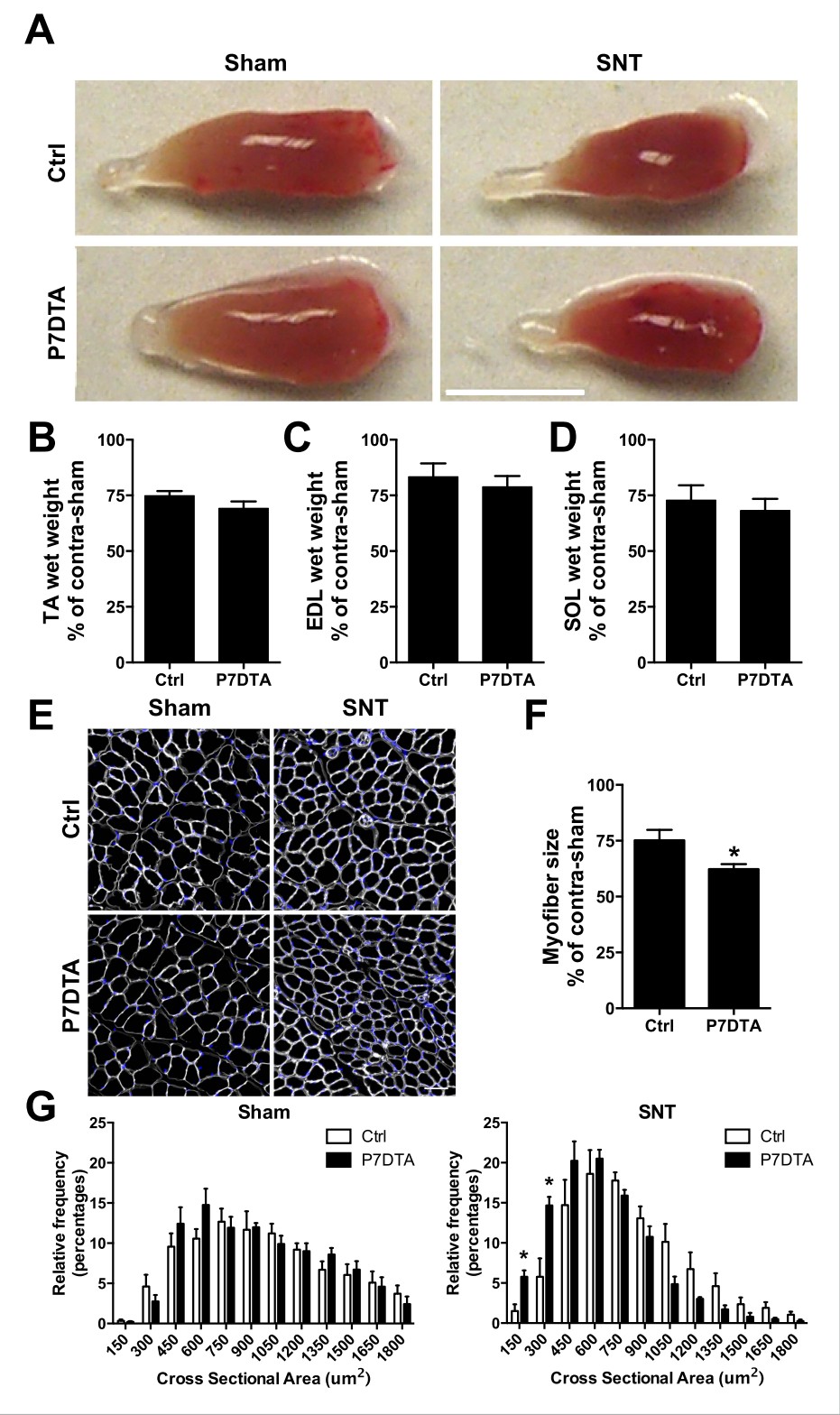

Figure 2. SC depletion exacerbates neuromuscular disruption induced myofiber atrophy. (A) Representative images of TA muscles. Scale bar = 5 mm. (B–D) Quantification of (B) TA (C) EDL and (D) Soleus (SOL) muscle wet weight after SNT as a percentage of contralateral sham. N = 6 for Ctrl and 8 for P7DTA. (E) Representative TA sections stained with anti-Laminin (white) and DAPI (blue). Scale bar = 50 μm. (F) Quantification of TA myofiber size as
*Figure 2. continued on next page*

*Figure 2. Continued*

a percentage of contralateral sham. (**G**) Histograms of TA myofiber size distributions. N = 4 mice, ≥1000 myofibers/mouse. *p < 0.05, t-tests.
The following figure supplement is available for figure 2:

**Figure supplement 1**. Retention of myofibers and modest loss of myonuclei 6 weeks after SNT.

antibodies specific for MyHC I, MyHC IIA and MyHC IIB revealed: i) very little if any expression of MyHC I; ii) inductions of MyHC IIA that were aggravated upon SC depletion; and iii) loss of MyHC IIB (*Figure 4A and C–E*). Together, these data indicate that the majority of hybrid myofibers observed after SNT are presumably type IIA/IIX.

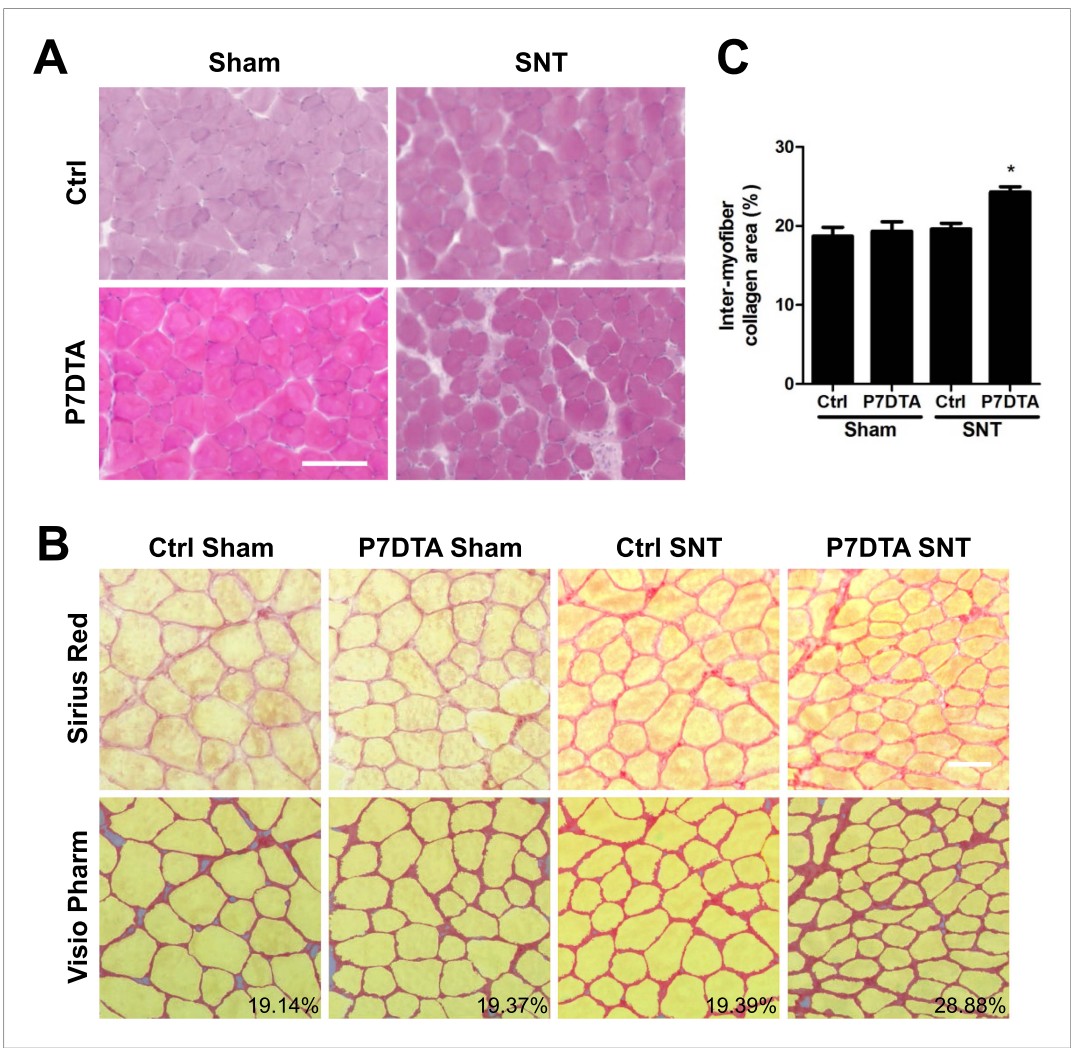

**Figure 3**. SC depletion induces connective tissue accumulation in skeletal muscles after neuromuscular disruption. (**A**) Representative images of TA sections stained with H&E; scale bar = 100 μm, (**B**) Representative images of TA sections stained Sirius Red and pseudocolor images generated by VisioPharm software; numbers indicate myofiber connective tissue (MCT) (red) content in each representative image; scale bar = 50 μm. (**C**) Quantification of MCT content in TA muscles. N = 4 mice. *p < 0.05 compared to Ctrl-sham, P7DTA-sham and Ctrl-SNT, ANOVA/Bonferroni multiple comparisons test.

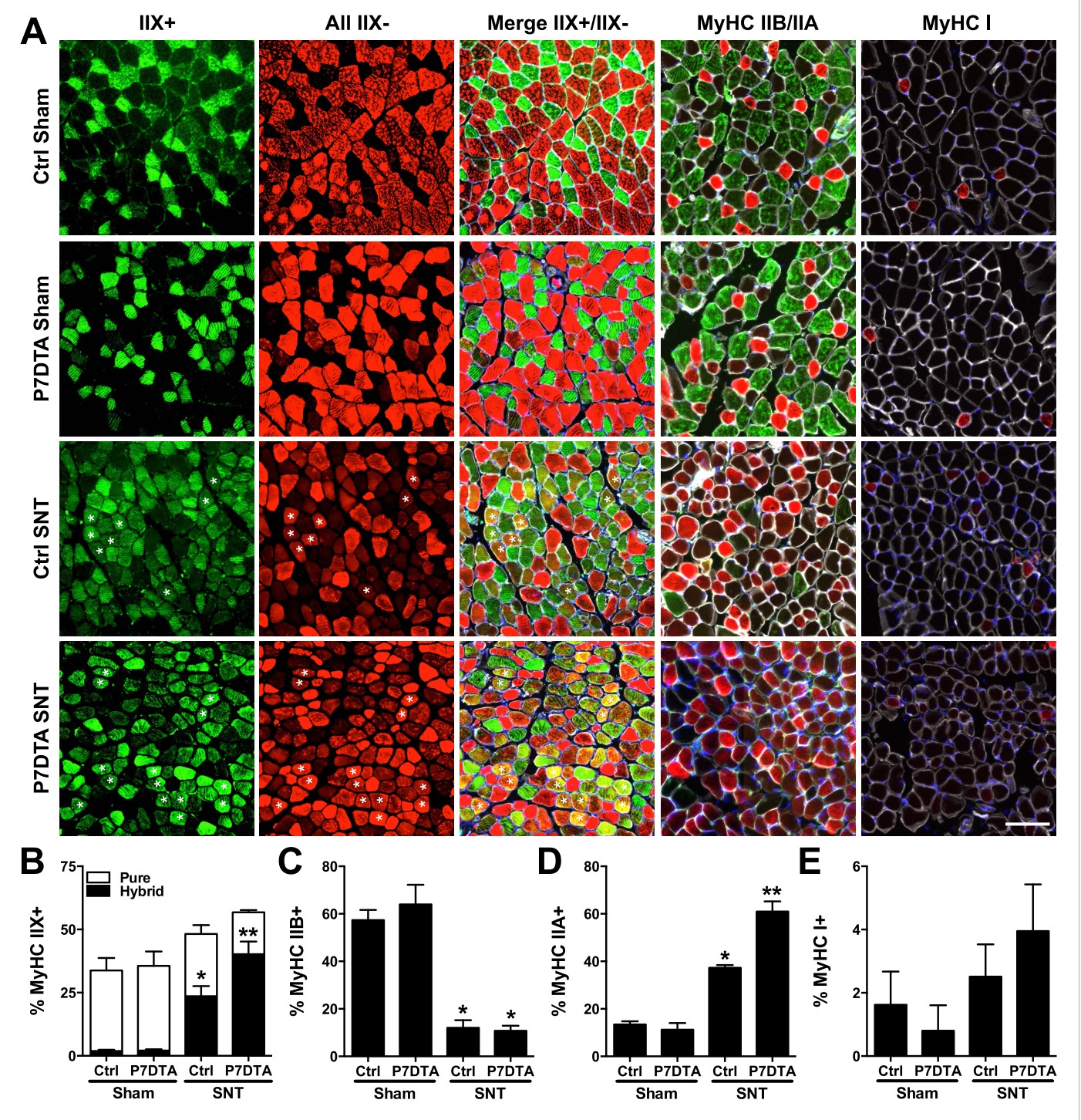

Figure 4. SC depletion aggravates myofiber type transitions connected to neuromuscular disruption. (A) Representative images of Ctrl and P7DTA inner TA/EDL muscle regions 6 weeks after sham and SNT surgery stained as indicated with anti-MyHC IIX, all MyHCs except IIX, MyHC IIA, MyHC IIB and MyHC I. Also depicted in Merge IIX+/IIX-, MyHC IIB/IIA and MyHC I labeled images are stains for anti-Laminin (white) and DAPI (blue). (B) Quantification of type IIX pure (green only) and hybrid (green and red, labeled with asterisks) myofiber percentages. (C–E) Quantification of (C) Type IIB (D) Type IIA and (E) Type I fiber percentage. N = 4 mice, 3 sections/mouse, 3 fields/section. Scale bar = 50 µm. *p < 0.05 compared to Ctrl-sham and P7DTA-sham, **p < 0.05 compared to Ctrl sham, P7DTA-sham and Ctrl-SNT, ANOVA/Bonferroni multiple comparisons test.

Modulations in factors that influence force transmission and excitation/contraction coupling, such as increased MCT and abnormalities in the expression of contractile elements can hinder intrinsic whole skeletal muscle contractile function (*Baldwin and Haddad, 2001*; *Kjaer, 2004*). Since we observed inductions in MCT and deviations in the expression of MyHC isoforms, major components of the contractile apparatus, after SC depletion and SNT, we measured *ex vivo* muscle contractility of Ctrl and P7DTA EDL muscles. Initially we examined the time taken by a given muscle to reach peak tension upon stimulation at 150 Hz, a frequency that elicited peak tetanic contractile force. A similar lengthening in time to peak tension (TTP) was observed in Ctrl and P7DTA EDL muscles after SNT (*Figure 5A*, and *Figure 5—figure supplement 1B*). Although lengthening of TTP 6 weeks after SNT is consistent with shifts in MyHC expression with slower contractile character, SC depletion did not further influence this property. Next we examined the max tetanic force generated by Ctrl and P7DTA EDLs 6 weeks after SNT surgery upon stimulation at frequencies of increasing intensity. We found significant deficits in peak absolute and specific force generation in P7DTA EDL muscles in comparison to Ctrl after SNT at progressively higher frequencies (*Figure 5B,C*, and *Figure 5—figure supplement 1A*). Therefore, the exacerbated deficits in skeletal muscle integrity such as myofiber atrophy, increased MCT content and abnormal profiles of MyHC expression upon SC depletion and SNT were accompanied by declines in whole skeletal muscle force generation.

## Neuromuscular disruption stimulates SC contribution in the vicinity of NMJs

Initially to assess SC fate we employed an in vivo BrdU incorporation assay to examine SNT-induced SC activation and division (*Chakkalakal et al., 2012*). Consistent with limited SC activation, only a fraction (~25%) of the SC pool in TA muscles had incorporated BrdU after SNT (*Figure 6A,B*). To further examine the fate of Pax7+ SCs in response to neuromuscular disruption, we generated a SC specific *Pax7^{CreER/+}*; *Rosa26^{mTmG/+}* (P7mTmG) transgenic mouse line. The P7mTmG mouse ubiquitously expresses a loxP flanked membrane tomato red fluorescent reporter (RFP) that undergoes Tmx mediated excision to indelibly label Pax7+ SCs and derived cells with membrane GFP (GFP), enabling lineage tracing of SCs in skeletal muscles and individual myofibers after SNT (*Murphy et al., 2014*). To initially characterize the efficiency of the P7mTmG line, we assessed GFP label in SCs and myofibers 24 hr after the last Tmx administration. We found that 24 hr after the last Tmx administration, >90% of the SC pool was GFP+, whereas myofibers were devoid of GFP label (*Figure 7—figure supplement 1*). Next, we examined SC derived GFP labeling after sham or SNT surgery. In comparison to sham, a marked increase in GFP + myofibers was observed in transverse sections from the mid belly of TA

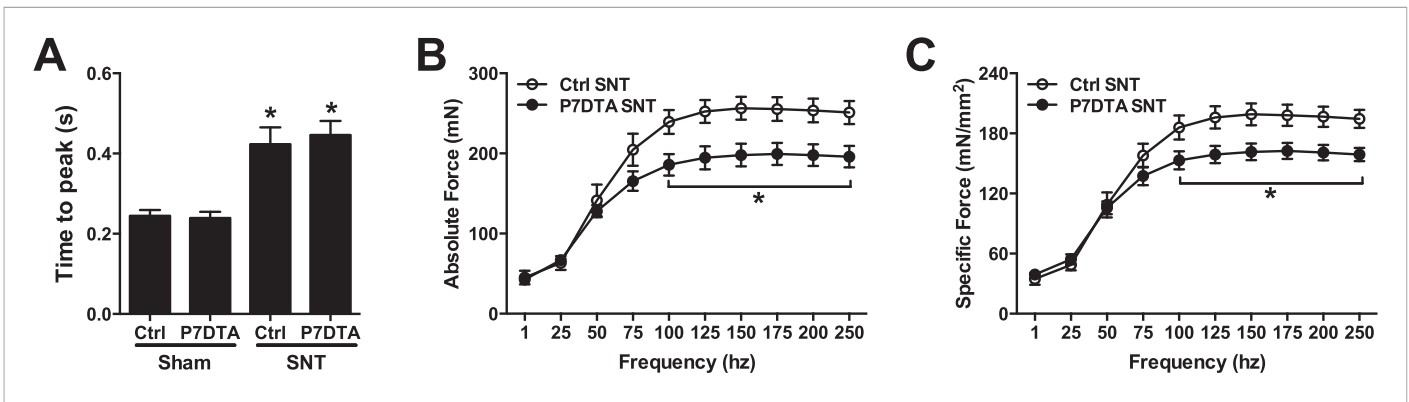

**Figure 5**. SC depletion leads to declines in force generation of skeletal muscles after neuromuscular disruption. (**A**) Average time to peak tension (TTP) during 150 Hz stimulation in EDL muscles. *p < 0.05 compared to Ctrl and P7DTA sham. ANOVA/Bonferroni multiple comparisons test, N = 4–6. (**B**) Absolute and (**C**) Specific force frequency curves for Ctrl and P7DTA EDL muscles 6 weeks after SNT surgery. *p < 0.05 compared to Ctrl SNT at indicated frequency, t-tests, N = 4–6 mice.

The following figure supplement is available for figure 5:

**Figure supplement 1**. Reduced contractile force and slowed force development in EDL muscles following SNT.

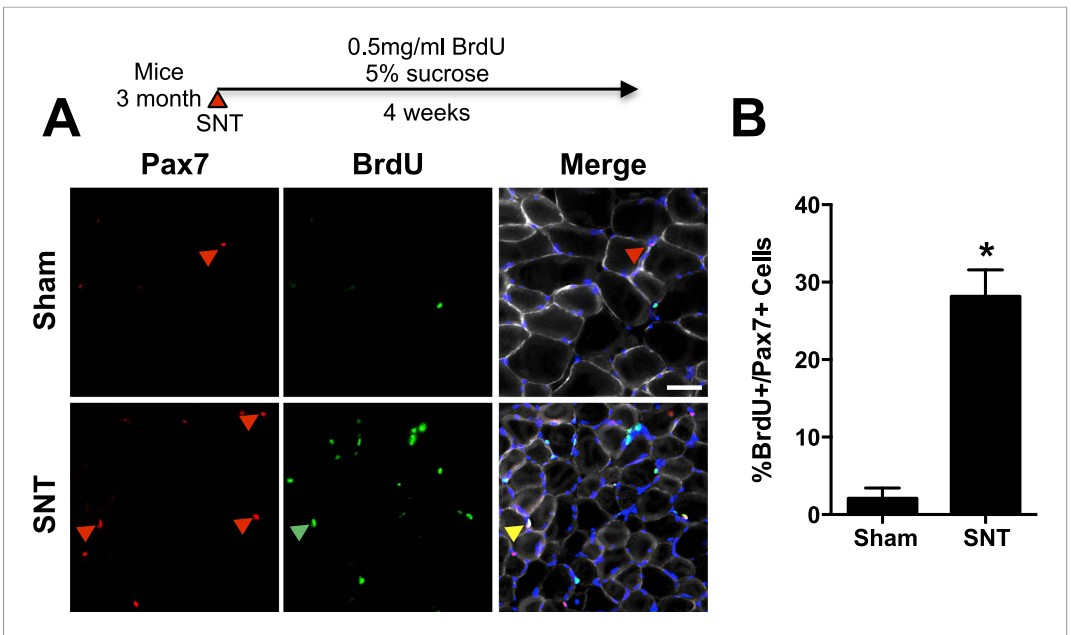

**Figure 6**. Limited SC proliferation in skeletal muscles upon neuromuscular disruption. (**A**) Strategy to BrdU label SCs in adult mice 4 weeks after sham or SNT surgery and representative TA sections stained with anti-Pax7 (red), anti-BrdU (green) and anti-Laminin (white). Red arrowheads indicate Pax7+ cell; green arrowhead indicates BrdU + cell; yellow arrowhead indicates BrdU+/Pax7+ cell. (**B**) Quantification of BrdU+/Pax7+ percentage. N = 3 mice, 3 sections/mouse, 6 fields of view/section. *p < 0.05, t-tests.

and EDL muscles after SNT (*Figure 7A,B*). Although these data indicate a high proportion of myofibers within TA and EDL muscles have undergone a fusion event after SNT, this activity likely reflects limited fusion within a given myofiber consistent with modest losses of myonuclei within P7DTA myofibers (*Figure 2—figure supplement 1A,C*). The occurrence of central nucleated myofibers (CNF), an indicator of degenerative/regenerative events of myofibers, was minimal and cannot explain the observed induction of GFP + myofibers after SNT (*Figure 7—figure supplement 1C*).

Myofibers are long multinucleated cells, along which local events of turnover and gene activity can occur (*Pavlath et al., 1989*; *Li et al., 2011*). To determine if SNT leads to regional activity and fusion of indelibly labeled SCs and derived progenitors, we isolated EDL single myofibers by collagenase digestion and processed them for the detection of post-synaptic acetylcholine receptors (AChRs) with fluorochrome-conjugated α-bungarotoxin (Btx) and GFP (*Chakkalakal et al., 2012*, *2014*). We scored the GFP labelling based on location: i) 'whole' for GFP throughout the myofibers; ii) 'end' for GFP at myofiber ends/tips; iii) 'middle' (Mid) for central regions of myofibers where NMJs are located; and iv) 'RFP' for no GFP labelling (examples for RFP, End and Mid in *Figure 7C,D*). After sham surgery, only a small fraction of myofibers were GFP positive; none of these fibers had GFP throughout, and most labels were found either at the ends or middle regions, the latter being where NMJs are located (*Figure 7E*). After SNT surgery a greater proportion of myofibers were GFP positive, indicating the activation of at least some SCs and fusion of SC derived progenitors to myofibers (*Figure 7E*). In addition, the majority of the GFP label found in myofibers isolated from EDL muscles after SNT was located in the middle, within the vicinity of NMJs (*Figure 7E*). Therefore, neuromuscular disruption led to regional SC derived contributions along the length of a myofiber in proximity to the NMJ.

## SC depletion impairs NMJ regeneration after neuromuscular disruption

Due to the regionalized response of SCs along myofibers near the NMJ after SNT, we employed P7DTA mice to test the importance of this local SC activity to NMJ regeneration upon neuromuscular disruption. Confocal IF microscopy and 3-D image analysis with Amira were used to assess NMJ regeneration. NMJs were identified with post-synaptic (AChR labeled with Btx) and pre-synaptic

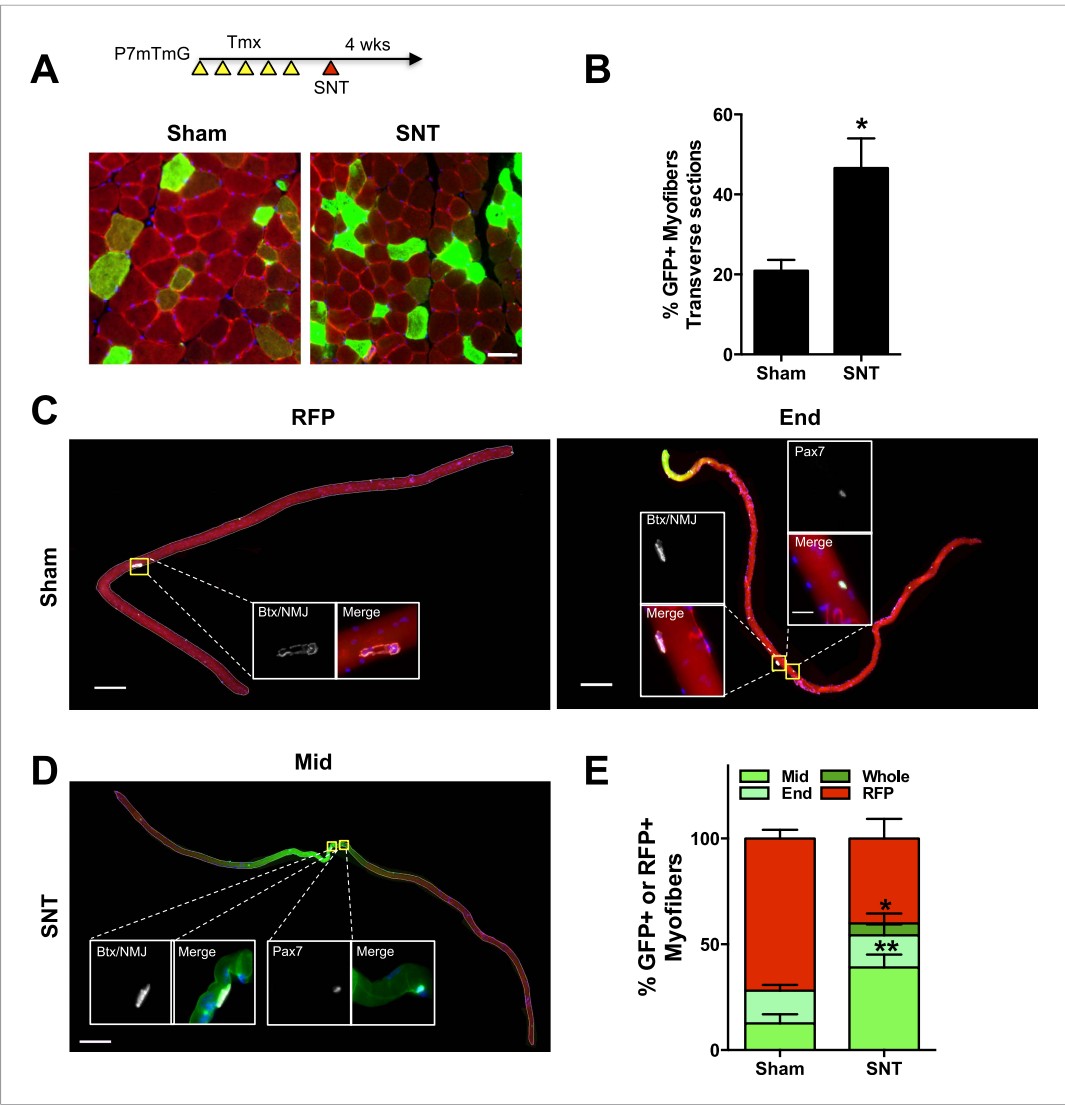

**Figure 7**. Neuromuscular disruption stimulates SC contribution in the vicinity of NMJs. (**A**) Scheme demonstrating time of tamoxifen treatment, SNT surgery, and harvest of tissue. Representative images to examine GFP label in myofibers and Pax7+ SCs of P7mTmG skeletal muscles 4 weeks after sham and SNT surgery. (**B**) Quantification of the percentage of GFP + myofibers from midbelly of EDL muscles 4 weeks after sham or SNT surgery. Scale bar = 50 μm. N = 3 mice, 3 sections/mouse, 6 fields/section. *p < 0.05, t-tests. (**C, D**) Representative images of single isolated P7mTmG EDL myofibers with no GFP (RFP), GFP at ends (End) or GFP in middle portions where neuromuscular junctions (NMJs) are located (Mid) after (**C**) sham or (**D**) SNT surgery. Magnified inset images show SCs (Pax7+) or NMJs (Btx, AChRs). Scale bar for myofibers = 200 μm for inset = 25 μm. (**E**) Quantification of GFP + fiber percentage and distribution. Note a higher percentage of myofibers after SNT express GFP primarily in the Mid regions, the location of NMJs. N = 4 mice, 25 myofibers examined per mouse. *p < 0.05 for all GFP + groups, **p < 0.05 for Mid GFP, t-tests.

The following figure supplement is available for figure 7:

**Figure supplement 1**. P7mTmG myofibers are not GFP + when examined immediately after Tmx administration.

---

(SV2, Syt-2 and neurofilament) markers (*Figure 8A*) (*Williams et al., 2009*; *Valdez et al., 2010*). Initially, we examined the extent of reinnervation, defined as the coverage of post-synaptic regions by pre-synaptic markers. We considered a NMJ to be: i) innervated, if the vast majority of post-synaptic regions are covered by pre-synaptic terminal markers; ii) partially denervated, if > 5 μm length of an

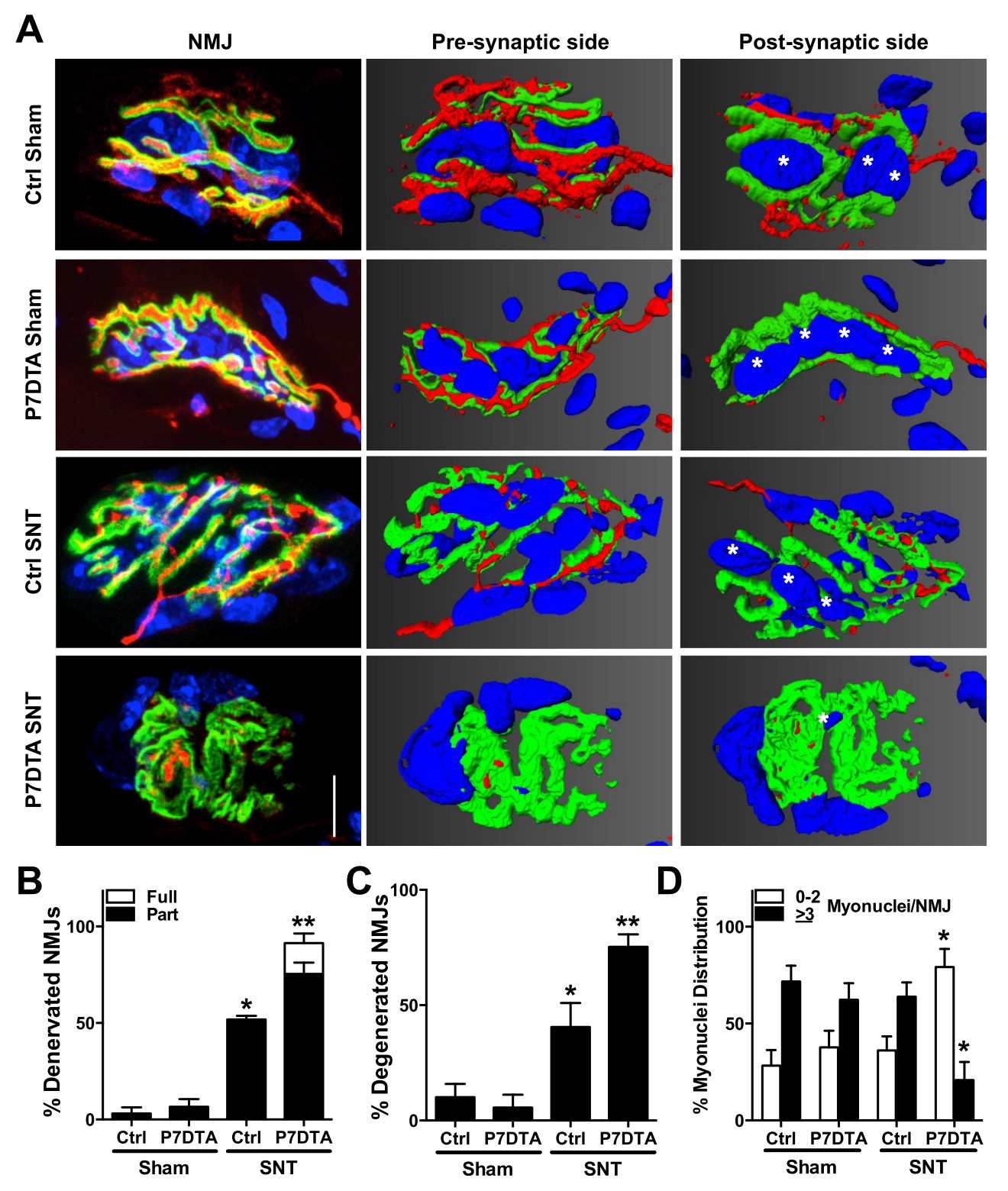

**Figure 8**. Reductions in NMJ reinnervation, post-synaptic morphology, and post-synaptic myonuclei in SC depleted skeletal muscle. (**A**) Representative confocal IF images and 3-D Amira based reconstructions of Ctrl and P7DTA NMJs 6 weeks after sham or SNT surgery, stained for post-synaptic (AChRs labeled with Btx, green), pre-synaptic markers (SV2, Syt-2, neurofilament, red) and myonuclei (DAPI, blue). Post-synaptic myonuclei are indicated with asterisks. (**B**) Quantification of NMJ reinnervation: partially dennervated (Part) and fully denervated (Full). (**C**) Quantification of degenerated NMJs based on post-synaptic morphology. (**D**) Percentage distribution of NMJs based on number of post-synaptic myonuclei. Scale bar = 10 μm. N = 4 mice,

*Figure 8. Continued*

20 NMJs/mouse. *p < 0.05 compared to Ctrl and P7DTA sham, **p < 0.05 compared to Ctrl-sham, P7DTA-sham and Ctrl-SNT, ANOVA/Bonferroni multiple comparisons test.
The following figure supplement is available for figure 8:

**Figure supplement 1**. Loss of post-synaptic myonuclei is a feature of NMJ degeneration.

AChR enriched branch within the post-synaptic apparatus is not covered by pre-synaptic terminal markers while the other parts of the apparatus are; and iii) fully denervated, if > 90% of the post-synaptic apparatus is devoid of pre-synaptic nerve terminal markers. Consistent with previous reports, a large increase in partially/fully denervated NMJs in Ctrl TA muscles was observed after SNT compared to sham surgery, suggesting incomplete regeneration of NMJs (*Figure 8B*) (*Williams et al., 2009*). Examination of P7DTA NMJs revealed that SC depletion was associated with a significantly higher proportion of NMJs that remained partially/fully denervated 6 weeks after SNT surgery (*Figure 8B*).

Post-natal NMJ maturation is associated with post-synaptic morphological transitions from a plaque-like to a pretzel-like shape due to the formation of post-synaptic membrane invaginations (junctional folds) (*Sanes and Lichtman, 1999*; *Wu et al., 2010*). Aged and NMD-afflicted skeletal muscles are frequently associated with NMJ degeneration, manifested by loss of elaborate AChR-enriched branches and the characteristic pretzel-like morphology of mature adult NMJs (*Valdez et al., 2010*, *2012*). Because post-synaptic morphology is linked to NMJ integrity and disease, we measured the occurrence of NMJ degeneration in Ctrl and P7DTA skeletal muscles after sham or SNT surgery. We considered an NMJ to be plaque-like, or degenerated, if the AChR-enriched area resembled a patch devoid of defined elaborate branches >5 µm in length. Very few degenerated NMJs were found in Ctrl or P7DTA TA muscles 6 weeks after sham surgery regardless of genotype (*Figure 8C*). After SNT surgery there was a marked increase in the proportion of degenerated NMJs, a percentage that was substantially elevated by ∼ 25% with SC depletion (*Figure 8C*).

Enriched at most adult NMJs are clusters of post-synaptic myonuclei specialized for the expression of genes required for AChR consolidation and the differentiation of pre-synaptic nerve terminals (*Sanes and Lichtman, 1999*; *Hippenmeyer et al., 2007*; *Zhang et al., 2007*; *Mejat et al., 2009*). Since SCs function as an essential source of myonuclei for skeletal muscle regeneration, local SC activity along denervated myofibers may be required for the maintenance of post-synaptic myonuclei (*Relaix and Zammit, 2012*). Therefore, we quantified the number of post-synaptic myonuclei (>25% DAPI covered by the Btx + post-synaptic apparatus) in TA muscles (*Grady et al., 2005*; *Zhang et al., 2007*; *Mejat et al., 2009*). Assessment of Ctrl and P7DTA NMJs 6 weeks after sham surgery did not reveal any significant alteration in the distribution of NMJs based on post-synaptic myonuclei size (*Figure 8D*). Also, no significant loss of post-synaptic myonuclei was found at Ctrl NMJs 6 weeks after SNT (*Figure 8D*). However, partially/fully denervated or degenerated (based on post-synaptic morphology, plaque-like) Ctrl NMJs had significantly fewer post-synaptic myonuclei, indicating that loss of post-synaptic myonuclei is a feature of NMJ degeneration (*Figure 8—figure supplement 1*). Consistent with a higher proportion of degenerated NMJs, a significant shift towards smaller post-synaptic myonuclear size was observed at P7DTA NMJs 6 weeks after SNT (*Figure 8D*). Collectively these data indicate that SC depletion, although not sufficient alone to trigger loss of NMJ integrity at homeostasis, did notably impair NMJ regeneration in response to neuromuscular disruption.

## Discussion

In this report we interrogated the roles and fates of SCs in a model of neuromuscular regeneration. In doing so we found that upon NMJ disruption: i) SC depletion exacerbates myofiber atrophy and type transitions; ii) SC depletion leads to elevated fibrosis and declines in muscle force generating capacity; iii) SC-derived contributions prevail in the vicinity of NMJs; and iv) SC depletion leads to deficits in skeletal muscle reinnervation, reductions in post-synaptic morphology and loss of post-synaptic myonuclei. Through these findings we propose a cellular mechanism whereby SCs contribute to the regeneration of NMJs and skeletal muscle maintenance upon neuromuscular disruption. Based on the

results presented here, an aspect of NMJ regeneration includes local SC activation, derived progenitor fusion and turnover of post-synaptic myonuclei.

Although we did not observe any major alterations in whole muscle mass after SNT, SC depletion did lead to further declines in myofiber size and increased MCT accumulation. The connection between SC depletion and MCT accumulation or fibrosis has also been observed by other studies in the context of skeletal muscle regeneration, functional overload-induced hypertrophy and aging (*Murphy et al., 2011*; *Fry et al., 2014*, *2015*). A study by the Kardon group highlighted that MCT fibroblasts are dynamically regulated by SCs, and the positive feedback between SCs and fibroblasts ensures efficient and effective skeletal muscle regeneration (*Murphy et al., 2011*). Results from the Peterson group revealed that SC depletion led to a fibrotic response in long-term overloaded muscles and elevated MCT accumulation in aged muscles, suggesting that SCs regulate the myofiber extracellular environment through inhibiting fibroblast function during skeletal muscle remodeling (*Fry et al., 2014*, *2015*; *Lee et al., 2015*). In addition, multiple populations of muscle interstitial cells (MICs) have been identified with fibro-adipogenic potential, such as Fibro-adipogenic progenitors (FAPs), PDGFRα+ MICs or ADAM12 + perivascular cells (*Joe et al., 2010*; *Uezumi et al., 2010*; *Dulauroy et al., 2012*; *Malecova and Puri, 2012*). These cells may serve as functional niche components for SCs, and their cross-talk with SCs may regulate muscle regeneration and fibrosis associated with NMDs (*Malecova and Puri, 2012*). How the functional interactions between SCs, fibroblasts and MICs affect fibrosis either through cell contact or the release of soluble factors requires further study.

Myofiber type transition and declines in force generation capacity are highly correlated with NMJ disruption (*Windisch et al., 1998*; *Wu et al., 2014*). Rodent models revealed that peripheral nerve lesions cause fast to slow myofiber transitions and a substantial increase in the proportion of type IIA myofibers (*Windisch et al., 1998*; *Chakkalakal et al., 2012*). A preferential loss of fast-type type IIB neuromuscular synapses has been reported in aged and NMD-afflicted muscles (*Frey et al., 2000*). Consistent with previous studies, we showed an induction of type IIA and loss of type IIB myofibers after SNT. This SNT-induced increase of type IIA myofibers was aggravated upon SC depletion, in line with our later finding that more P7DTA NMJs remain denervated after SNT. Concomitantly, after SNT and SC depletion we observed a notable induction of hybrid myofibers, a typical feature of NMD-afflicted and aged skeletal muscle (*Klitgaard et al., 1990*; *Krivickas et al., 2002*; *Palencia et al., 2005*; *Stevens et al., 2008*; *Rowan et al., 2012*). The mechanisms responsible for the occurrence of MyHC coexpression are still elusive. One possibility is induction of multiple MyHC isoforms following denervation and reinnervation by multiple motor neurons that specify distinct myofiber types (*Pette and Staron, 2000*). Our results also revealed that SC depletion led to declines in force generation capacity after neuromuscular disruption. Both accumulated MCT content and abnormal expression of contractile proteins could partially account for the loss of whole skeletal muscle force generation after SC depletion and SNT (*Baldwin and Haddad, 2001*; *Kjaer, 2004*). Also, impaired NMJ regeneration in response to neuromuscular disruption might contribute to the additional loss of force generating capacity in SC-depleted muscles compared to Ctrl after SNT. In support, declines in force generating capacity are a feature of partially denervated and aged skeletal muscles (*Urbanchek et al., 2001*; *Kalliainen et al., 2002*).

Limited if any cellular turnover and no net loss of myonuclei has been reported in chronically denervated skeletal muscle (*Bruusgaard and Gundersen, 2008*). Therefore, it has been concluded that muscle mass loss in the context of denervation-induced atrophy primarily reflects imbalances between protein synthesis and degradation as opposed to cellular turnover (*Gundersen and Bruusgaard, 2008*). Our SNT surgery, however, leads to complete denervation and the eventual reinnervation of the vast majority of myofibers 4–6 weeks after surgery (*Buti et al., 1996*; *Williams et al., 2009*). Chronic denervation models do have greater levels of myofiber atrophy than observed with the SNT employed here (*Snow, 1983*; *Bruusgaard and Gundersen, 2008*). Although chronic denervation leads to SC activation, it is generally accepted that limited fusion of SC derived progenitors to atrophying parent myofibers occurs (*Borisov et al., 2005*; *Bruusgaard and Gundersen, 2008*; *Gundersen and Bruusgaard, 2008*). Rather, SCs and derived progenitors tend to migrate into interstitial spaces where they undergo apoptosis, are believed to contribute to the formation of nascent myofibers with distinct basal lamina juxtaposed to parent atrophying myofibers, or contribute to other forms of failed myogenesis (*Borisov et al., 2005*; *Bruusgaard and Gundersen, 2008*). It will be of interest to determine whether SCs are a source of nascent myofibers and failed myogenesis, and how this contributes to myofiber size changes during chronic denervation.

Even though only a small percentage of SCs were found to activate after SNT, surprisingly we found that SC activation and SC-derived progenitor fusion primarily occurred in central portions of myofibers, in the vicinity of NMJs. Moreover, more denervated and degenerated (based on post-synaptic morphology) NMJs observed in SC depleted skeletal muscles after SNT were associated with significant declines in the number of post-synaptic myonuclei at NMJs. These phenotypes indicate the importance of regionalized SC activity for regenerating NMJs. Central nucleation was not a prominent feature of myofibers 4–6 weeks after SNT. Recently transient central nuclei were observed along the length of aging myofibers at random locations (*Li et al., 2011*). Whether such transient degenerative regenerative events occur in the vicinity of NMJs in response to SNT as part of program of myogenic progenitor differentiation and fusion remains to be determined. The factors that control SC activity and derived progenitor fusion at central myofiber regions in the vicinity of NMJs during NMJ disruption are unknown. One possibility is the local expression of SC fate regulators that support progenitor activity and fusion from myofibers, the principal SC niche cell (*Bischoff, 1990*; *Yin et al., 2013*). Alternatively, denervation may trigger the expression of factors that suppress SC activation and differentiation at extra-synaptic myofiber regions. Factors implicated in SC quiescence and activation include loss of Notch signaling and increased fibroblast growth factor (FGF) or hepatocyte growth factor (HGF)-induced receptor tyrosine kinase signaling (*Yin et al., 2013*). TGFβ superfamily signaling, which is elevated in denervated skeletal muscle, is a well-established suppressor of myogenic differentiation (*Kollias and McDermott, 2008*; *Sartori et al., 2014*). In addition, Jak/Stat and Wnt signaling are factors that regulate SC derived myogenic progenitor differentiation (*Yin et al., 2013*; *Price et al., 2014*; *Tierney et al., 2014*). Whether any of these factors limit SC activation and differentiation in the vicinity of denervated NMJs will require molecular dissection of myofibers at distinct regions.

Post-synaptic myonuclei are specialized for the expression of synapse-enriched genes required for the development, differentiation, consolidation and maintenance of both pre- and post-synaptic components (*Schaeffer et al., 2001*; *Hippenmeyer et al., 2007*). Consistent with vital roles, the etiology of some NMDs includes reductions in post-synaptic myonuclear number and integrity. Emery-Dreifuss muscular dystrophy is characterized by a loss of post-synaptic myonuclei together with deficits in skeletal muscle innervation, post-synaptic AChR morphology and the induction of gene expression programs consistent with denervation (*Mejat et al., 2009*). A feature of Slow-channel syndrome, a congenital myasthenia disorder, includes apoptotic activity and the accumulation of DNA damage at post-synaptic myonuclei (*Zhu et al., 2014*). Genetic studies have also shown the importance of post-synaptic myonuclei towards NMJ development. Mice null for Syne-1, a nuclear anchoring protein, display loss of post-synaptic myonuclei together with gross deficits in the innervation of embryonic skeletal muscles (*Zhang et al., 2007*). Therefore, reductions of post-synaptic myonuclei in SC-depleted skeletal muscles after SNT could manifest in the loss of gene expression programs required for the regeneration and maintenance of NMJs.

Similar to what we observed with SC-depleted skeletal muscles after NMJ disruption, myofiber type transitions, decreases in NMJ regenerative capacity and force generation are also features of aged and NMD-afflicted skeletal muscles (*Frey et al., 2000*; *Verdu et al., 2000*; *Urbanchek et al., 2001*; *Kalliainen et al., 2002*; *Hegedus et al., 2008*; *Rowan et al., 2012*; *Kang and Lichtman, 2013*). Since loss of SC number and function is also associated with aging and NMDs, it will be of interest to determine in these contexts the interrelationship between reduced integrity of Pax7+ SCs impaired NMJ regeneration and correlated skeletal muscle dysfunction (*Pradat et al., 2011*; *Chakkalakal et al., 2012*; *Hayhurst et al., 2012*).

## Materials and methods

### Animals

C57BL/6, *Pax7CreERT2* (017763) *Rosa26mTmG* (007576) and *Rosa26DTA* (009669) mice were obtained from Jackson Laboratories (Bar Harbor, ME). *Rosa26mTmG* or *Rosa26DTA* mice were crossed with *Pax7CreERT2* mice to generate *Pax7CreER/+*; *Rosa26mTmG/+* (P7mTmG) or *Pax7CreER/+*; *Rosa26DTA/+* (P7DTA) mice and control CreER negative (Ctrl) littermates. Transgenic mouse lines were used at 3–6 months of age. P7DTA and Ctrl mice were treated with (0.1 mg Tmx/g body weight) for 5 consecutive days I.P., 7 days after the first Tmx injection mice underwent SNT and sham surgeries. P7DTA and Ctrl mice were given additional Tmx 10 and 17 days after surgery. P7mTmG mice were similarly administered Tmx,

however without additional Tmx injections after surgery. All animal procedures were conducted in accordance with institutional guidelines approved by the University Committee on Animal Recourses, University of Rochester Medical Center.

## Sciatic nerve transection (SNT)

Mice were anesthetized with intraperitoneal injections of ketamine (110 mg/kg) and xylazine (10 mg/kg). The hindquarter was then carefully shaved and depilation completed with generic Nair hair removal cream prior to skin cleansing with gauze. The skin was incised 1 mm posterior and parallel to the femur, and the biceps femoris was bluntly split to expose the sciatic nerve. 1–2 mm sciatic nerve was then transected 5 mm proximal to its trifurcation, followed with realignment of the distal and proximal nerve ends and closure of the muscle with wound clips (Autoclip, BD Clay Adams, Franklin Lakes, NJ). Mice were given analgesic (0.5–1.0 mg/kg buprenorphine) and allowed to recover on a heating pad. Sham surgery was performed on the contralateral leg where procedures were performed without nerve transection. At the designated times left and right hind limb muscles were collected. The muscles used for histology were incubated at 4°C overnight in 30% sucrose solution and frozen in dry ice-cooled isopentane.

## Immunofluorescence (IF)

Flash-frozen muscles were sectioned at 10 μm (transverse) or 30 μm (longitudinal) and stored at −80°C. Sections were fixed for 3 min in 4% paraformaldehyde (PFA) (no PFA fixation for MyHC antibodies), and if needed, subjected to antigen retrieval: heating slides in citrate buffer (10 mM sodium citrate, pH 6.0) in a steamer (Oster 6.1 quart, model 5712, Racine, WI) for 15 min followed by cooling at room temperature for 2 min (Tang et al., 2007). Tissue sections were incubated with 0.2% Triton X-100 for 10 min, blocked in 10% normal goat serum (NGS, Jackson ImmunoResearch, West Grove, PA) 30 min at room temperature and stained with primary antibodies. If necessary (when mouse primary antibodies were used), sections were blocked in 3% affinipure Fab fragment goat anti-mouse IgG(H + L) (Jackson ImmunoResearch, West Grove, PA 115-007-003) with 2% NGS at room temperature for 1 hr. Primary antibodies were incubated at 4°C overnight or 2 hr at room temperature, and secondary antibodies were incubated for 1 hr at room temperature. DAPI (Invitrogen, Carlsbad, CA) staining was used to mark nuclei. All slides were mounted with Fluoromount-G (SouthernBiotech, Birmingham, AL).

## Histology and bright-field microscopy

For H & E staining, flash-frozen sections were fixed for 3 min in 4% PFA, stained with Mayers Hematoxylin and Alcoholic Eosin Y, dehydrated, equilibrated with xylene and mounted using Cytoseal 60 (Richard-Allan Scientific, Kalamazoo, MI). For Sirius Red staining, a Picrosirius Red stain kit (Polysciences, Warrington, PA) was utilized. Briefly flash-frozen sections were fixed for 1 hr at 56°C in Bouin's fixative, washed in water, stained for 1 hr in Picrosirius Red, washed in 1 M HCl, dehydrated, equilibrated and mounted. Bright-field images were collected by a Zeiss Axioskop 40 microscope. Olympus VS110 virtual microscopy system was utilized for whole-slide scanning. Automatic quantification of MCT content was accomplished via VisioPharm software.

## Conventional and confocal IF microscopy and analysis

Transverse sections and cells were imaged on a Zeiss Axio Observer A.1 microscope. Longitudinal sections were stained with SV2, Znp-1, 2H3, Btx and DAPI as described above and viewed with an Olympus Fluoview 1000 confocal microscope with 40X (for quantification) or 100X (for representative pictures) objectives at a 0.47 μm or 0.42 μm step size respectively. Amira software was used to analyze 3-D reconstructed NMJs for innervation analysis and to identify and enumerate post-synaptic myonuclei. Max-projection z-stack images of NMJs were generated with ImageJ software. The post-synaptic side was identified based on the entry of the terminal axon and as the concave side of the NMJ.

## Single myofiber analysis

For GFP localization, myofibers were purified by conventional collagenase digestion and trituration with fire polished glass pipets as previously described (Zammit et al., 2004). Briefly, the EDL muscle was dissected, rinsed in Dulbecco's phosphate-buffered saline (PBS), put into a 1.5 ml eppendorf tube containing 1 ml 0.1% type I collagenase (Invitrogen) and 0.1% type II collagenase (Invitrogen) in Dulbecco's modified Eagles medium (DMEM, Sigma–Aldrich, St. Louis, MO), incubated in a shaker

water bath at 37°C for 75 min and gently mixed by inversion periodically. Following digestion, the muscle was transferred to 100 mm × 15 mm plastic petri dishes containing 10 ml of plating media (10% horse serum in DMEM) using fire-polished-tip Pasteur pipettes. Under a stereo dissecting microscope, single myofibers were released by gently triturating the EDL with a series of modified Pasteur pipettes that varied in tip diameter to accommodate the progressive decrease in muscle trunk size. Inseparable fibers and debris were removed. Purified single myofibers were fixed with 4% PFA for 3 min, washed with PBS and transferred to 5 ml polystyrene cell collection tubes for GFP, Pax7 and Btx IF. For assessing single myofiber size and myonuclear number, muscles were fixed in 4% PFA for 48 hr prior to dissection and NaOH mediated digestion(*Brack et al., 2005*). Fixed muscles were incubated in 40% NaOH for 2 hr and agitated vigorously for 20 min. Released fibers were then washed in PBS and processed for DAPI staining.

### In vivo cell division analysis

To assess cell proliferation C57BL/6 mice were fed BrdU (Sigma–Aldrich) (0.5 mg/ml supplemented with 5% sucrose) in drinking water after denervation (*Chakkalakal et al., 2012*). Muscles were collected 4 weeks after SNT and sectioned for Pax7, BrdU and Laminin IF.

### Ex vivo muscle contraction assay

Whole EDL muscle contractility and force generation were analyzed using an ASI muscle contraction system (Aurora Scientific, Aurora, Canada) as described previously (*Wei-Lapierre et al., 2013*). Briefly, mice were anaesthetized and TA muscles removed. EDLs were then carefully isolated, adjusted to optimal length and stimulated at various frequencies to obtain absolute force values. To obtain specific force values, absolute force was normalized to pennation angle and cross-sectional area (determined by EDL weight and length). Muscle force was recorded and analyzed using Dynamic Muscle Control, Clamp fit and Graph Pad Prism software.

### Antibodies

Pax7 (mouse IgG1, 1:100, Developmental Studies Hybridoma Bank (DSHB), Iowa City, IA), BrdU (rat, 1:250, Abcam ab6326, Cambridge, UK), Laminin (rat or rabbit, 1:1500, Sigma–Aldrich L0663 or L9393), GFP (rabbit, 1:400, Millipore AB3030P, Billerica, MA), F1.652 (mouse IgG1, 1:40, DSHB), A4.840 (mouse IgM, 1:40, DSHB), NCL-MHC (mouse IgG1, 1:100, DSHB), SC-71 (mouse IgG1, 1:40, DSHB), 6H1 (mouse IgM, 1:40, DSHB), BF-35 (mouse IgG1, 1:40, DSHB), BF-F3 (mouse IgG1, 1:40, DSHB), SV2 (synaptic vesicle protein-2, mouse IgG1, 1:100, DSHB), Znp-1 (synaptotagmin-2, mouse IgG1, 1:200, DSHB) and 2H3 (neurofilament, mouse IgG1, 1:200, DSHB), AlexaFluor 488-conjugated α-Bungarotoxin (1:1000, Life Technologies B-13422, Grand Island, NY), AlexaFluor 647-conjugated α-Bungarotoxin (1:1000, Life Technologies B-35450), AlexaFluor 594-conjugated goat anti-mouse IgG (1:1500, Life Technologies A-11032), AlexaFluor 594-conjugated goat anti-mouse IgG1 (1:1500, Life Technologies A-21125), AlexaFluor 488-conjugated goat anti-mouse IgM (1:1500, Life Technologies A-21042), AlexaFluor 488-conjugated goat anti-rat IgG (1:1500, Life Technologies A-11006), AlexaFluor 647-conjugated goat anti-rabbit (1:1500, Life Technologies A-21244).

### Data analysis

Results are presented as mean +SEM. Statistical significance was determined by Student's *t*-tests for simple comparison, one-way ANOVA and Bonferroni multiple comparisons test for multiple comparisons with Graph Pad Prism software. $p < 0.05$ was considered as statistically significant.

### Acknowledgements

We thank Dr. Paivi Jordan (University of Rochester Medical Center Confocal and Conventional Microscopy Core), Dr. Jonathan Carroll-Nellenback (University of Rochester Center for Integrated Research Computing), Dr. David Yule, Andrew Soroka, Erica Hange, and Center for Musculoskeletal Research Histology, Biochemistry, and Molecular Imaging Core for provision of tools, technical assistance and helpful reading of the manuscript. This work was supported by URMC start-up funds, DoD grant (W81XWH-14-1-0454), NIH grant (AG051456) (J.V.C), NIH grant (AR059646) (R.T.D), NYSTEM Training grant (C026877) (W.L) and NIH grant (1S10RR027340-01) (University of Rochester Medical Center, Center for Musculoskeletal Research: Histology, Biochemistry and Molecular Imaging Core).

## Additional information

### Funding

| Funder | Grant reference | Author |
|---|---|---|
| Congressionally Directed Medical Research Programs (CDMRP) | W81XWH-14-1-0454 | Joe V Chakkalakal |
| National Institute of Arthritis and Musculoskeletal and Skin Diseases (NIAMS) | R01AR059646 | Robert T Dirksen |
| NIH/NIA | R01AG051456 | Joe V Chakkalakal |
| NYSTEM | University of Rochester Training Grant C026877 | Wenxuan Liu |

The funders had no role in study design, data collection and interpretation, or the decision to submit the work for publication.

### Author contributions

WL, Conception and design, Acquisition of data, Analysis and interpretation of data, Drafting or revising the article; LW-LP, AK, Acquisition of data, Analysis and interpretation of data; RTD, Analysis and interpretation of data, Drafting or revising the article; JVC, Conception and design, Analysis and interpretation of data, Drafting or revising the article

### Ethics

Animal experimentation: This study was performed in strict accordance with the recommendations in the Guide for the Care and Use of Laboratory Animals of the National Institutes of Health. Work with mice was conducted in accordance with protocols approved by the University Committee on Animal Resources, University of Rochester Medical Center protocol(#101565/2013-002).

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
