## [Decision Letter]

Thank you for submitting your work entitled “Inducible depletion of adult skeletal muscle stem cells impairs the regeneration of neuromuscular junctions” for peer review at *eLife*. Your submission has been favorably evaluated by Sean Morrison (Senior Editor), Amy Wagers (Reviewing Editor), and two reviewers.

The reviewers have discussed the reviews with one another and we have drafted this decision to help you prepare a revised submission.

All the reviewers found your manuscript to be well written, with clear and compelling data using state-of-the-art techniques. Using inducible depletion of muscle satellite cells in vivo, the work shows convincingly that there are significant effects of satellite cell loss on myofiber size, accumulation of fibrotic tissue, myonuclear number, NMJ regeneration and muscle contractile activity in the context of experimentally induced denervation. Given ongoing discussions in the field regarding the contribution of satellite cells to these processes, the work is both timely and important, and likely to be of broad interest. The reviewers did have a few suggestions and queries to clarify certain points of data analysis and interpretation, which should be addressed on revision.

Essential revisions:

1) Given that mice were only followed for 6 weeks after satellite cell depletion, concluding that the modest decline in satellite cell numbers with aging contributes to NMJ impairments and muscle dysfunction associated with aging is overly speculative. Following mice for a longer period of time, into old age, in the absence of nerve transection is required to determine if satellite cells play a role in normal maintenance of the NMJ. Please temper these conclusions or provide additional data (if available).

2) There appears to be a disconnection in the quantification of different morphological features of the muscles. Nerve transection results in a decrease in myonuclear number only in control mice, not in P7DTA mice, and yet approximately 60% of fibers appear to have undergone satellite cell fusion in the lineage tracing experiment (Figure 7). This apparent discrepancy should be discussed.

3) The representative images shown in Figure 7 do not match the quantification in Figure 7. In particular, far less than 20% of fibers appear to be labeled in Sham mice, which seems unlikely over the course of only 4 weeks. Similarly, the images shown in Figure 3 do not reflect that 20% of the muscle area is occupied by collagen in Sham mice (Figure 3). Please address these apparent discrepancies.

4) Given that nearly 100% of NMJs are at least partially denervated and 75% degenerated at 6 weeks in P7DTA mice, the rather modest effect on fiber size is surprising. Comparison to mice, 6 weeks following nerve transection in which regeneration is totally blocked, would be helpful for perspective and to judge the relative impact of satellite cell depletion on the process of nerve regeneration. Presumably those data are available in the authors' lab or in the literature.

5) The data regarding the occurrence of centrally nucleated fibers are a bit confusing as presented (in the subsection “Neuromuscular disruption stimulates SC contribution in the vicinity of NMJs”). It would be helpful to present these in graphical format. Also, are the authors indicating that the contribution of satellite cells to myofibers that is spurred by denervation does not produce central nucleation? Or that the incorporated nuclei are localizing specifically to the synaptic region? Some discussion in the text on this point would be useful.

---

## [Author Response]

1) Given that mice were only followed for 6 weeks after satellite cell depletion, concluding that the modest decline in satellite cell numbers with aging contributes to NMJ impairments and muscle dysfunction associated with aging is overly speculative. Following mice for a longer period of time, into old age, in the absence of nerve transection is required to determine if satellite cells play a role in normal maintenance of the NMJ. Please temper these conclusions or provide additional data (if available).

As requested we have modified our statements with regards to aging and satellite cell decline in the Discussion.

*2) There appears to be a disconnection in the quantification of different morphological features of the muscles. Nerve transection results in a decrease in myonuclear number only in control mice, not in P7DTA mice, and yet approximately 60% of fibers appear to have undergone satellite cell fusion in the lineage tracing experiment (*Figure 7*). This apparent discrepancy should be discussed.*

There is a significant difference after SNT in P7DTA mice; we have modified the legend for Figure 2—figure supplement 1 to correct this omission. Regardless, we agree that the magnitude of myonuclei loss was modest in response to SNT in P7DTA muscles. We believe this reflects in part the moderate loss of myonuclei in P7DTA sham myofibers. We have included some sentences for this observation (in the subsection “SC depletion exacerbates myofiber atrophy and induces connective tissue accumulation in skeletal muscles after neuromuscular disruption”). We apologize if we have given the impression that there is a direct correlation between the proportion of myofibers within a given muscle that have undergone a fusion event (in theory 1 out of hundreds of myonuclei within a myofiber), and myonuclear number within an individual myofiber. We have included a statement to clarify that myofiber proportions within an entire skeletal muscle should not be correlated with myonuclei loss within an individual myofiber (in the subsection “Neuromuscular disruption stimulates SC contribution in the vicinity of NMJs”).

*3) The representative images shown in*
Figure 7
*do not match the quantification in*
Figure 7*. In particular, far less than 20% of fibers appear to be labeled in Sham mice, which seems unlikely over the course of only 4 weeks. Similarly, the images shown in*
Figure 3
*do not reflect that 20% of the muscle area is occupied by collagen in Sham mice (*Figure 3*). Please address these apparent discrepancies.*

We have provided representative images that better reflect our quantification in Figure 7. The quantification of picrosirius red label for collagen was done with VisioPharm software. VisoPharm obtains numbers from pseudo-colored images that enable greater sensitivity of detection. Therefore we have also included the pseudo-colored images and representative numbers in Figure 3. Accordingly, the legend for Figure 3 has been modified.

4) Given that nearly 100% of NMJs are at least partially denervated and 75% degenerated at 6 weeks in P7DTA mice, the rather modest effect on fiber size is surprising. Comparison to mice, 6 weeks following nerve transection in which regeneration is totally blocked, would be helpful for perspective and to judge the relative impact of satellite cell depletion on the process of nerve regeneration. Presumably those data are available in the authors' lab or in the literature.

Unfortunately we do not have data in models where nerve regeneration is totally blocked (chronic denervation). We have included a discussion about myofiber size declines observed in our study to data obtained in other studies from the literature where nerve regeneration was totally blocked (Discussion).

5) The data regarding the occurrence of centrally nucleated fibers are a bit confusing as presented (subsection “Neuromuscular disruption stimulates SC contribution in the vicinity of NMJs”). It would be helpful to present these in graphical format. Also, are the authors indicating that the contribution of satellite cells to myofibers that is spurred by denervation does not produce central nucleation? Or that the incorporated nuclei are localizing specifically to the synaptic region? Some discussion in the text on this point would be useful.

The data regarding central nucleated myofibers are now presented as a graph in Figure 7—figure supplement 1. As requested we have included some discussion as to how SC derived myogenic progenitors may contribute to NMJs (Discussion).